# Role of GPCR Signaling in Anthracycline-Induced Cardiotoxicity

**DOI:** 10.3390/cells14030169

**Published:** 2025-01-22

**Authors:** Nimish Biswal, Ritika Harish, Minahil Roshan, Sathvik Samudrala, Xuanmao Jiao, Richard G. Pestell, Anthony W. Ashton

**Affiliations:** 1School of Medicine, Xavier University at Aruba, Oranjestad, Arubaxuanmao.jiao@bblumberg.org (X.J.); richard.pestell@bblumberg.org (R.G.P.); 2Pennsylvania Cancer and Regenerative Medicine Research Center, Baruch S. Blumberg Institute, Wynnewood, PA 19096, USA; ritika.harish@bblumberg.org; 3The Wistar Institute, Philadelphia, PA 19104, USA; 4Division of Perinatal Research, Kolling Institute of Medical Research, University of Sydney, St Leonards, NSW 2065, Australia; 5Division of Cardiovascular Medicine, Lankenau Institute for Medical Research, Wynnewood, PA 19096, USA

**Keywords:** doxorubicin cardiotoxicity, RNA sequencing, GPCR signaling

## Abstract

Anthracyclines are a class of chemotherapeutics commonly used to treat a range of cancers. Despite success in improving cancer survival rates, anthracyclines have dose-limiting cardiotoxicity that prevents more widespread clinical utility. Currently, the therapeutic options for these patients are limited to the iron-chelating agent dexrazoxane, the only FDA-approved drug for anthracycline cardiotoxicity. However, the clinical use of dexrazoxane has failed to replicate expectations from preclinical studies. A limited list of GPCRs have been identified as pathogenic in anthracycline-induced cardiotoxicity, including receptors (frizzled, adrenoreceptors, angiotensin II receptors) previously implicated in cardiac remodeling in other pathologies. The RNA sequencing of iPSC-derived cardiac myocytes from patients has increased our understanding of the pathogenic mechanisms driving cardiotoxicity. These data identified changes in the expression of novel GPCRs, heterotrimeric G proteins, and the regulatory pathways that govern downstream signaling. This review will capitalize on insights from these experiments to explain aspects of disease pathogenesis and cardiac remodeling. These data provide a cornucopia of possible unexplored potential pathways by which we can reduce the cardiotoxic side effects, without compromising the anti-cancer effects, of doxorubicin and provide new therapeutic options to improve the recovery and quality of life for patients undergoing chemotherapy.

## 1. Introduction

Over the last 30 years, the mortality associated with a cancer diagnosis has decreased 29% from its peak in 1991 [1], representing 2.9 million additional lives saved. Currently, ~16.9 million people are living with a diagnosis of cancer, and this is projected to increase to 21.7 million by 2029 [2]. The improved survival of cancer patients, with more than one-third surviving at least 5 years after their initial diagnosis, is due in part to the use of chemo- and radio-therapies. The acute and long-term toxicities of cancer treatments contribute increasingly to morbidity and mortality in cancer survivors [3], primarily through significant cardiotoxic side effects that diminish quality of life. The main chemotherapeutic agents exhibiting cardiotoxicity are anthracyclines (doxorubicin, daunorubicin, and epirubicin), ErbB2 inhibitors (trastuzumab), TKIs (sunitinib and sorafenib), proteasome inhibitors (carfilzomib), and immune-based therapies (immune checkpoint inhibitors) [4]. Considering breast cancer alone, currently >1.5 million women develop breast cancer annually. Approximately 30% of these patients receive doxorubicin at the recommended dose (<550 mg/m^2^) [5]. Under these conditions >50,000 women/year will develop severe cardiotoxicity in response to therapy. With cancer survivors estimated at 19 million in the USA by 2025 [6], cancer-therapy-induced cardiotoxicity is considered a “cardio-oncology epidemic” currently without effective diagnostics and therapies.

Gprotein-coupled receptors (GPCRs) are integral membrane proteins, with a seven-transmembrane α-helixes, that represent the largest receptor superfamily in the human genome with >800 protein-coding genes [7]. More than 100 GPCRs have been identified in cardiac tissues and many have been implicated in cardiac homeostasis, remodeling, and the progression of heart failure. GPCR-modulating therapeutics are particularly successful in the treatment of cardiovascular pathologies, including heart failure, hypertension, and arrhythmia [8]. However, the contribution of GPCR signaling to the cardiotoxic side effects of cancer therapy remains underappreciated. Given their effectiveness in other cardiac syndromes, the exploration of GPCR signaling in cardiotoxicity is long overdue and may reveal new avenues for improved cardiovascular therapeutics for an iatrogenic condition with few viable treatment options. This review will provide an overview of the mechanisms, and role of GPCR signaling, of chemotherapy-induced cardiotoxicity, and benefits of targeting GPCR signaling to minimize therapy-associated cardiotoxicity.

## 2. Epidemiology of Cardiotoxicity

Cardiotoxicity from cancer therapy manifests as a range of conditions including asymptomatic left ventricular dysfunction, congestive heart failure (HF), myocardial ischemia, myocarditis, QT prolongation, and arrhythmia [9]. However, the largest controversy centers on the definition of cardiomyopathy and HF. The first definition of cancer-therapy-induced cardiotoxicity was based on serial decline in left ventricular ejection fraction (LVEF) [10]. The most recent definition of cardiotoxicity from the American Society of Echocardiography includes a LVEF drop of >10% to an absolute value of <50%, whilst the definition of left ventricular dysfunction (LVD) is broader: “a decrease in cardiac LVEF that was either global or more severe in the septum; symptoms of congestive heart failure (CHF); associated signs of CHF, including but not limited to S3 gallop, tachycardia, or both; and decline in LVEF of at least 5% to less than 55% with accompanying signs or symptoms of CHF, or a decline in LVEF of at least 10% to below 55% without accompanying signs or symptoms” [11].

Cancer-therapy-induced cardiotoxicity can broadly be divided into irreversible (type 1) and reversible (type 2) damage, although there is contention over the reversible nature of type 2 (Table 1) [10]. Type 1 cardiotoxicity has a greater association with cardiac dysfunction and clinical heart failure, whilst type 2 shows more reversibility and is associated with increased loss of contractility and less myocyte death. Most drugs inducing type 2 injury promote vascular dysfunction and ATP depletion in myocytes, resulting in cell loss [10]. Indeed, HER2 expression on cardiac myocytes is important for protection from cardiotoxins and its attenuation by trastuzumab results in cardiotoxicity. The cardiotoxic effects of therapies producing type 2 injury can be treated with the discontinuation of therapy and/or standard medications for heart failure (ACE inhibitors, calcium channel blockers, β-blockers). Moreover, the cardiotoxic side effects of many of the above agents (5-FU, taxanes, trastuzumab, radiation) are exacerbated when in combination with anthracycline therapy [12]. Given the dramatic synergy on LVEF and the irreversible nature of anthracycline cardiotoxicity, we will focus on the cardiotoxic effects of this chemotherapeutic for the remainder of this review.

**Table 1 cells-14-00169-t001:** **Different classes of cardiotoxicity with diverse anti-cancer therapies.** HF—heart failure; Hyper—hypertension; TKI—tyrosine kinase inhibitor; FU—fluorouracil; VEGF—vascular endothelial growth factor.

Type	Class	Predisposition	Anti-Chemotherapeutic Agents	Cardiac Complication	Ref.
Type 1	Irreversible	Cumulative dose	Anthracyclines (daunorubicin, doxorubicin, epirubicin, idarubicin	9–48% HF Arrythmia	[13,14,15,16]
			Alkylating agents (busulfan, carboplatin, carmustine, chlormethine, cisplatin, cyclophosphamide, mitomycin)	7–28% HF	
			Taxanes (docetaxel, cabazitaxel, paclitaxel); topoisomerase inhibitors (etoposide, tretinoin, vinca alkaloids)	2.3–8% HF	
			Antimetabolites (cladribine, cytarabine, 5-FU)	1.6–20% HF	
Type 2	Reversible	Not related to cumulative dose	ErbB2 antagonists (trastuzumab, lapatinib)	3–7% HF	[17]
			VEGF antagonists (bevacizumab)	4–35%Hyper	
			TKIs (sorafanib, sunitinib)	17–47%Hyper	

**Figure 1 cells-14-00169-f001:**
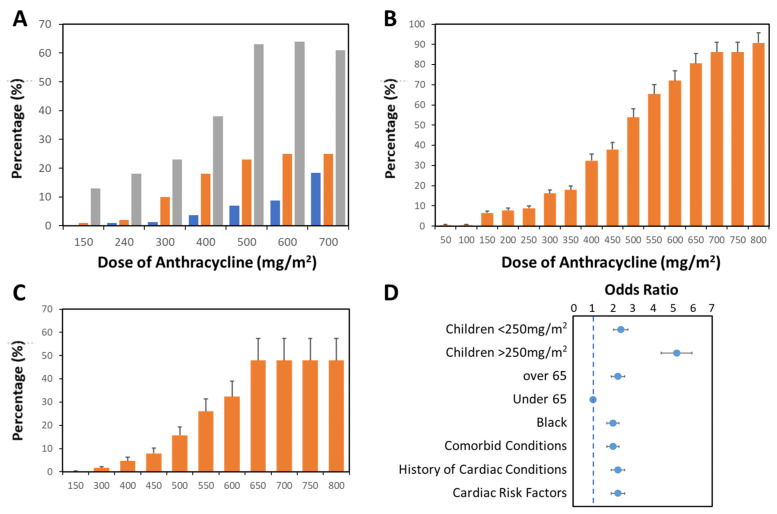
**Risk of cardiotoxicity increases with dose of DOX used in cancer patients.** Cardiotoxicity was assessed in cancer patients receiving DOX for primary management of the indicated tumor. (**A**) Incidence of acute (■), delayed (5–10 years, ■), and chronic (>10 years, ■) cardiotoxicity is elevated with escalating DOX dose in cancer patients. (**B**,**C**) When stratified by DOX dose, the incidence of cardiotoxicity (**B**; loss of EF to ≥20% or reduction in EF by 10% from baseline) and heart failure (**C**) in breast cancer patients (n = 630) becomes clear. (**D**) Odds ratio for DOX cardiotoxicity in a Non-Hodgkins Lymphoma patient population (n = 141) [17]. Data (mean ± SEM) are from [18,19,20].

The anthracycline class of chemotherapeutic drugs includes doxorubicin, daunorubicin, epirubicin, and idarubicin. Among these, doxorubicin (DOX) stands out as the most widely used anthracycline in the effective treatment of many cancers, including lymphoma, myeloma, breast, ovarian, lung, gastric, thyroid, sarcoma, and pediatric cancers [21]. Doxorubicin is the most cardiotoxic drug of the class, with epirubicin and idarubicin being 0.7 and 0.53 times as toxic [19]. Of patients treated with anthracyclines, ~10% experience acute cardiotoxicity (as defined above) within the first year following completion of chemotherapy; however, this figure increases to 25% over a 5-year period (Figure 1A) [22]. DOX produces severe, dose-dependent cardiotoxicity (LVEF < 50%) in 9% (250 mg/m^2^), 18% (350 mg/m^2^), and 38% (450 mg/m^2^) of patients affected within 12 months of cessation of therapy (Figure 1B) and up to 60% of patients affected over 10 years at doses at or above 500 mg/m^2^ (Figure 1A). Moreover, the risk of congestive heart failure increases proportionally to the cumulative dose of DOX (5% chance at 400 mg/m^2^ and 48% at 650 mg/m^2^) (Figure 1C), resulting in clinical use being capped at ~450 mg/m^2^. Recovery is also related to dose, with no recovery of LVEF in patients receiving >346 mg/m^2^ of DOX, whilst those receiving doses between 314 and 346 mg/m^2^ show partial recovery. Other than lifetime cumulative dose, the risk factors for susceptibility to DOX-induced cardiotoxicity include being very young (>2.5-fold increase) or very old (over 65, >2-fold increase), having co-morbid conditions, especially cardiovascular risk factors (>2-fold increase) and Black ethnicity (>2-fold increase) (Figure 1D) [18,19].

## 3. Mechanisms of DOX-Induced Cardiotoxicity

Multiple pathways are modulated by anthracyclines to manifest cardiotoxicity. As a drug class, anthracyclines bind to DNA and topoisomerase (Top)IIb (Figure 2) with blocking TopIIb being the classic mechanism of action of toxicity in cardiac cells. Two cellular activities following TopIIb poisoning in myocytes are linked to the pathogenesis of anthracycline cardiotoxicity: (1) DNA damage (caused by double-strand breaks (DSBs)) [23,24] and (2) chromatin damage (mediated via histone eviction) (Figure 2). TopII unwinds supercoiled DNA allowing for transcription and replication. DOX and other anthracyclines intercalate directly into DNA to inactivate TopII. Anthracycline treatment promotes the supercoiling and, ultimately, the overwinding of DNA, producing double-strand breaks [25,26]. Since TopIIb is inactivated, it cannot re-ligate the DNA compaction [27]. Additionally, the increased DNA damage activates the TP53 system, causing cell cycle arrest and, ultimately, cell death [28]. The involvement of TopIIb in DOX cardiotoxicity was unequivocally demonstrated in mice after the cardiomyocyte-specific deletion of Top2b reduced DNA double-strand breaks, transcriptome changes, and defective mitochondrial biogenesis and ROS formation [29].

The other effects of intercalation of anthracyclines into dsDNA is to ultimately compete with histones for space (Figure 2). Eventually, histones are displaced from bound DNA at genomic regions marked by H3K36me3 [30] and the nucleosome (DNA–histone complex) collapses. The arrangement of chromatin is important for many processes including transcription and replication. Once evicted, histones are replaced with nascent chromatin that is inherently less organized. The newly associated histones have different epigenetic modifications and result in chromatin damage [31]. Anthracycline-induced cardiotoxicity requires both DNA- and chromatin-damaging activities [32], whilst anti-tumor activity requires only chromatin damage. Indeed, the engineering of DOX variants to minimize DNA double-strand break activity preserves anti-cancer functions and ablates cardiotoxic side effects in murine and human models [32]. These results suggest that anthracycline variants that primarily exert their effects through chromatin damage may preserve cardiac function and quality of life for cancer patients.

Indirect effects of anthracyclines include structural changes to rRNA, impaired ribosomal function, and activating the ribosomal/ribotoxic stress response [33]. This promotes a cascade of events [29] including nucleolar stress with the suppression of new pre-ribosomal RNA synthesis [33], the induction of p70RSK, the activation of autophagy [34,35], the inhibition of mammalian Target of Rapamycin (mTOR) [36], the disruption mitochondrial electron transport [37,38,39] and cardiac ion channels [40,41], oxidative stress from mitochondrial iron accumulation and lipid peroxidation in mitochondria [39], the activation of proteases [42], the induction of cellular senescence [43], and myocardial cell death (apoptosis, ferroptosis, and necrosis) [32]. The ATP dependence of cardiac myocytes makes the heart especially sensitive to mitochondrial damage [44]. Anthracyclines promote both intrinsic and extrinsic pathways of apoptosis and upregulates the expression of cardiomyocyte cell surface membrane death receptors (such as the cell surface death receptors Fas, DR4, DR5, and TNF receptor 1 (TNFR1)) [45] and intrinsic pathways (BAX translocation, increased p53 stability) in vivo [45] and in isolated cardiomyocytes in vitro [46,47,48]. The major driver seems to be excessive ROS production, consequent to TopIIb inactivation, and the recruitment/oxidation of cardiolipin [47]. The deletion of TopIIb in the heart abolished DOX-induced cardiac toxicity [49], demonstrating the central role of TopIIb inhibition.

The activation of both JNK and p38 MAPK promote the release of inflammatory cytokines, chemokines, and growth factors (IL1α, TNF-α, IL-6, CXCL1/Gro-α, CCL2/MCP-1, granulocyte colony stimulating factor (GCSF), and CXCL10/IP-10) from bone-marrow-derived macrophages [50]. DOX enhances the activity of natural killer cells and cytotoxic T-lymphocytes, and the differentiation of macrophages. The cancer therapeutic efficacy of anthracyclines involves the induction of M1 and the downregulation of M2 macrophages [51] and the accumulation of immature macrophages (CD11b + F4/80 + Gr-1 (Ly6C/Ly6G+) [52]) and antigen-presenting cells (APC) exhibiting a dendritic cell-like (i.e., CD11b + .CD11c. + Ly6ChighLy6G-_MHCH+) phenotype and T-cell subsets [53,54]. These DOX-induced immune responses contribute to the activation of a caspase cascade involving Toll-like receptors (TLRs) [46].

## 4. Existing Therapies Are Ineffective for Anthracycline Cardiotoxicity

Approaches to the treatment or prevention of anthracycline-induced cardiotoxicity showed that antioxidants are ineffective [55,56]. Additional approaches including liposomal reformulations of DOX (which preferentially enters into the leaky microvasculature of tumors) limits extravasation into cardiomyocytes; however, the increased risk of mucositis and hand–foot syndrome has led to the use of these formulations primarily in the setting of refractory or metastatic disease [55]. The chelation of free iron with deferoxamine is also ineffective in protecting against anthracycline cardiotoxicity in preclinical models [52]. In contrast, dexrazoxane (dexrazoxane hydrochloride), which also chelates intracellular iron, inhibits iron-assisted oxidative radical production and reduces the formation of superoxide radicals and consequent DNA damage [57]. Dexrazoxane is currently the only USFDA-approved drug to prevent DOX-induced cardiomyopathy in patients [58]. Its use has been limited for patients with metastatic breast cancer who have received a cumulative lifetime dose of ≥300 mg/m^2^ of DOX or an equivalent dose of other anthracyclines [59]. However, dexrazoxane has the possibility [54] of reducing the efficacy of the anthracycline and increases the risk of myelotoxicity [53], and is therefore not used routinely. Experimental preclinical studies have shown that PI3K inhibition [60], ginsenoside Rg3 micelles [61], statins [62], and PARP inhibitors [63] reduce DOX-induced cardiotoxicity; however, these agents have not yet transitioned to clinical use and new alternatives to reduce the cardiotoxicity of anthracyclines are urgently needed. Effective therapies for treating anthracycline-induced cardiotoxicity need to either exploit tissue-specific differences between cancerous tissues and the cardiomyocyte/coronary endothelium, or, more specifically, affect the cardiotoxic mechanisms without disrupting anti-tumor pathways.

## 5. G Protein-Coupled Receptors Are Regulated at Many Levels

G protein-coupled receptors (GPCR) are the largest family of cell membrane proteins and are responsible for physiologic actions in every type of tissue [56]. GPCRs consist of seven hydrophobic-transmembrane-spanning domains connected by intracellular (ICL1, ICL2, and ICL3) and extracellular (ECL1, ECL2, and ECL3) loops (Figure 2). GPCR activation is accompanied by the outward movement of transmembrane helices V and VI, which creates a docking site for heterotrimeric G proteins [64,65,66,67,68,69,70,71]. Agonist-activated GPCRs act as guanyl nucleotide exchange factors (GEFs) that facilitate the release of GDP and the binding of GTP to the Gα-subunit of heterotrimeric G proteins [72] (Figure 2). The GTP-ligated Gα-subunit undergoes conformational change and dissociates from the receptor and βγ-subunit, whereupon both G protein subunits bind their respective effectors (such as PKA, cAMP) (Figure 2). The vacated active GPCR is then free to initiate another round of Gα-GTP loading, which provides signal amplification.

A counterpoint to this process is the limitation of GPCR signaling. Gα signaling is spontaneously terminated as a consequence of the intrinsic GTPase activity of the Gα-subunit, which hydrolyzes bound GTP to GDP, resulting in the reformation of the inactive G protein heterotrimer. The intrinsic GTPase activity of α-subunits is generally insufficient to correlate with the physiological rates of G protein inactivation; however, this activity is accelerated by GTPase-activating proteins (GAPs), such as the regulator of G protein-signaling (RGS) proteins (Figure 2). Moreover, the phosphorylation of GPCRs by G protein-coupled receptor kinases (GRKs) and recruitment of β-arrestins (Figure 2) to GPCRs prevents further heterotrimeric G protein recruitment, desensitizes the receptor to the ligand, and induces receptor internalization and ubiquitination, resulting in receptor destruction/recycling (Figure 2). Thus, the contributions of GPCRs to cardiac homeostasis are regulated at many levels and the perturbation of any of these pathways could result in phenotypic changes capable of inducing pathology, such as cardiotoxicity in response to anthracyclines. We will now explore the established roles of GPCR signaling in anthracycline-induced cardiotoxicity and the future avenues for clinical translation.

## 6. GPCRs Are Significant Contributors to the Pathogenesis of Anthracycline Cardiotoxicity

There has been significant interest in establishing the pathological mediators of DOX-induced cardiotoxicity. Significantly, many of these mediators are ligands for GPCRs with the activation of multiple GPCRs protective against DOX-induced cardiotoxicity (Table 2). Melatonin inhibits necrosis and apoptosis, and improves cardiac dysfunction, through antioxidant effects and the suppression of lipid peroxidation without compromising the anti-tumor effect of DOX in mice and rats [68,69,70,73]. Ghrelin’s cardioprotective effects manifest through blocking AMPK and activating p38-MAPK pathways [74,75,76]. The co-administration of the GalR1-3 agonist [RAla14, His15]-galanin prevents the increase in plasma CK-MB activity, improves the parameters of cardiac function, and causes weight gain in acute models of DOX-mediated cardiac injury in rats [77,78].

The expression of the apelin receptor protein, APJ, is lost from the myocardium with acute DOX treatment, and the genetic deletion of APJ in mice increased the severity of DOX-induced heart injury, including impaired contractile function and survival [79,80]. Conversely, apelin protects H9c2 cardiomyocytes overexpressing APJ against DOX-mediated cell death [79], suggesting that the loss of APJ signaling is an essential step in developing the myopathy. Similarly, prokineticin activation (with non-peptide agonist IS20) inhibits DOX-mediated cardiotoxicity in cultured cardiomyocytes and endothelial cells as well as mouse models of acute and chronic cardiotoxicity [81]. Importantly, these small molecules do not alter the cytotoxic effects of DOX in cancer cells and in in vivo cancer-cell-line-derived xenograft mouse models [78]. The intravenous pretreatment of mice with an adenosine receptor agonist (Cl-IB-MECA) ameliorates DOX-induced bradycardia, circulating creatine kinase, histologic evidence of cardiac remodeling, and oxidative stress injury [82,83]. ADORA3 activation significantly inhibited DOX-induced inflammatory responses (NFκB and TNFα expression) and prevented cardiac myocyte apoptosis as measured by cytochrome c release.Thus, targeting the loss of multiple GPCRs/ligands in preclinical models of DOX-mediated cardiotoxicity promotes myocyte health and reduces cardiac remodeling.

The endocannabinoid system has been investigated for both its cardioprotective and cardio-depressive effects. A primary prevention study in DOX-treated rats, showed improved cardiac function and left ventricular wall thickness after pretreatment with anandamide [84]. Cannabidiol protects the heart against DOX-induced cardiac injury in rats [85] and mice [86] by (i) attenuating ROS/RNS accumulation, (ii) preserving mitochondrial function and biogenesis, (iii) promoting cell survival, and (iv) decreasing myocardial inflammation. However, these results contradict the effects of the FAAH knockout mouse where elevated anandamide levels were correlated with depressed cardiac function and elevated ROS and mortality in DOX-induced cardiomyopathy [87]. 

**Table 2 cells-14-00169-t002:** **Cardioprotective GPCRs identified in models DOX cardiotoxicity.** Gα—heterotrimeric G protein α-subunit; Wnt—wingless-related integration site; CBD—cannabidiol; CPK—creatine phosphokinase; NFAT—nuclear factor of activated T-cells; SERCA—sarcoplasmic/endoplasmic reticulum Ca-ATPase; LV—left ventricle; TNF—tumor necrosis factor; IL—interleukin.

GPCR	Gα Coupling	Mechanism of Protection in DOXTOX Models	Reference
Adrenergic Receptor (α1-AR)	Gαq	Dabuzalgron (α1-AR agonist) reduces ROS production and fibrosis, preserves mitochondrial function and ATP levels, and augments contractile function in neonatal rat myocytes	[88,89]
Melatonin Receptor (MTNR1A/B)	MTRN1A-Gαi MTRN1B-Gαs	Melatonin diminished serum CPK elevation, reduced myocyte membrane damage, reversed impaired contractility, decreased coronary flow, and decreased heart rate in mice and rats. Effects due to antioxidant regulation and suppression of lipid peroxidation	[73,90,91,92]
Prokineticin Receptor (PKR1)	Gαq/11	PKR1 agonist (IS20) reduced apoptosis, ROS generation, necrosis, and collagen deposition/fibrosis in H9c2 rat cardiomyocytes. PKR1 activation augmented detoxification and cardiac output	[81,93]
Ghrelin Receptor (GSHR)	Gαq	GSHR activation reduced pyknotic nuclei and nuclear levels of NFAT-4. Simultaneously, administration of acetylated Ghrelin increased left ventricular systolic pressure and SERCA2a activity, improved mitochondrial function, restored muscle fibers, and improved myocyte ultrastructure	[75,76]
Galanin Receptor (GalR1/R2)	GalR1-Gαi/Gαo GalR2-Gαq	In male Wistar rats, a modified galanin fragment decreased LV systolic dysfunction, LV remodeling, and plasma CK-MB levels and increased ATP	[94,95,96]
Adenosine Receptor (ADORA3)	Gαi/Gαo	A3-adenosine receptor agonist preserved normal cardiac structure, decreased oxidative stress (increased reduced glutathione) and lipid peroxidation, and increased antioxidant enzymes in Wistar rats	[82,83]
(Endo)Cannabinoid Receptors (CB1 and CB2)	CB1-Gαi/Gαo	CBD, when used in C57BL/6J mice, attenuated cardiac dysfunction, oxidative stress, and myocardial lipid peroxidation, while enhancing mitochondrial biogenesis in damaged hearts. CB1 activation attenuated myocardial expression of TNF-α and IL-1β and decreased myocyte death	[84,85,86,87,97,98,99]
Apelin Receptor (APJ)	Gαi/o	APJ and apelin are lost from the myocardium and circulation, respectively, with acute DOX treatment in mice. APJ deletion exacerbates disease	[79,80]
Wnt Receptors (FZD1-10)	Multiple	Dickkopf (Dkk)1 was significantly increased (>3-fold) in DOX-treated myocytes in vitro and in vivo. Overexpression/exogenous Dkk1 decreased Bcl2 expression and promoted apoptosis in DOX-treated H9c2 cells. Moreover, viral delivery of Dkk1 to the myocardium exacerbated cardiac remodeling in response to DOX. These data suggest that the effects of Wnt signaling is cardioprotective in doxorubicin cardiotoxicity	[100,101]

Moreover, CB1 activation is deleterious in human primary cardiac myocytes treated with DOX in vitro, perhaps through the dysregulation of pathways involving reactive oxygen/nitrogen species [98]. This is supported by the finding that a CB1 antagonist (SR141716A or AM281) improved DOX-induced cardiac dysfunction and cardiomyocyte apoptosis, indicating that abrogation of CB1 signaling might prevent cardiac toxicity [97].

Wnt signaling takes place through a complicated network of GPCRs (frizzled 1–10) and accessory receptors (LRP5,:LRP6, Ror2, Drl, Ryk), is antagonized by Dickkopf (Dkk), and secretes frizzled-related proteins (sFRP1) [102,103]. Acute DOX challenge in rats abrogates Wnt/β-catenin signaling in the myocardium; however, Wnt/PCP-JNK signaling was activated [101]. The inhibitor of canonical Wnt signaling Dkk1 is increased (>3-fold) in DOX-treated myocytes in vitro and in vivo [100]. Dkk1 overexpression decreased Bcl2 expression and promoted apoptosis. Moreover, the viral delivery of Dkk1 to the myocardium exacerbated cardiac remodeling in response to DOX [100]. Conversely, the expression of sFRP1 (inhibitor of both canonical and non-canonical Wnt signaling) is lost upon DOX treatment in vitro and is associated with the loss of cell viability [101]. Pretreatment with an inhibitor of Wnt/PCP-JNK signaling and sFRP1 overexpression attenuated DOX-induced apoptosis [101]. Thus, preserving canonical Wnt/β-catenin signaling may be a therapeutic strategy in DOX-induced cardiotoxicity.

Interestingly, whether a particular mediator will be cardioprotective or cardiotoxic may be determined by the cadre of the receptor isoforms present. A fitting example of this are the adrenergic receptors Table 2 and Table 3). Recent studies suggested that dabuzalgron reduced ROS and fibrosis, enhanced contractile function, and preserved myocardial ATP content in DOX-treated mice [88]. The cardioprotective effects of dabuzalgron/α1-AR activation are attributed to its antioxidant and anti-apoptotic properties, achieved through MAPK1/2 [88], and A Kinase Anchoring Protein-Lbc (AKAP-Lbc)/protein kinase D1 activation [89], rather than its β-AR blocking activity, because carvedilol inhibits mitochondrial complex-I that promotes cardiotoxicity [104]. This cardioprotective effect of carvedilol is superior to metoprolol and atenolol for preventing DOX-induced apoptosis.

Like β-adrenergic receptors, other GPCR ligands are cardiotoxic and promote DOX-mediated injury (Table 3). In addition to its well-known functions of mediating hypertension, cardiac hypertrophy, heart failure, and fibrosis, the angiotensin (AT)II type 1 receptor also promotes the loss of histone 2A phosphorylation and increased histone mono-methylation to promote chromatin damage through the heterochromatin associated protein, HP1α, suggesting a role for ATIIR1 in the chromatin damage associated with DOX therapy [105]. Significantly, ATII type 1α receptor-null mice and the AT1 antagonist (RNH-6270) attenuated the DOX-induced loss of cardiac function and structural remodeling in acute and chronic models of cardiotoxicity [106].

The neuropeptide substance P and its cognate high-affinity receptor neurokinin 1 (NK-1R) are both elevated in the myocardium in response to chronic DOX therapy [107] (Table 3). NK-1R antagonists at low doses abrogate changes in cardiac function in DOX-treated mice and normalize markers of apoptosis, oxidative stress, and hypertrophy [107,108], suggesting that enhanced substance P signaling is a key mediator of DOX-induced cardiotoxicity, although new data may bring this into question [109]. Substance P release is enhanced from breast cancer cells by DOX, and NK-1R antagonists enhance cell death and ROS production in MDA-MB231 TNBC cells [108], suggesting that substance P antagonism as an adjunct therapy may protect the heart and augment the anti-tumor activity of DOX.

Cardiac μ-opioid receptor expression is enhanced by DOX (3.2- and 6.1-fold for RNA and protein, respectively) and temporally associated with the loss of cardiac function [110] (Table 3). Morphine mimics promote loss in left ventricular-dp/dt and increase oxidative stress and myocardial damage (Troponin T release) [111]. Morphine enhances the cardiotoxicity of DOX, likely through the enhanced cardiac μ-opioid receptor expression associated with acute DOX treatment (3.2- and 6.1-fold for RNA and protein, respectively) [110]. Finally, a non-selective opioid receptor antagonist ablated the detrimental effects of DOX on cardiac function in an acute model of cardiotoxicity [111], implicating endogenous opiate signaling in DOX cardiotoxicity.

**Table 3 cells-14-00169-t003:** **Cardiotoxic GPCR signaling on DOX cardiotoxicity.** Gα—heterotrimeric G protein α-subunit; ATII—Angiotensin II, μOR—mu Opioid Receptor; GPR-G-portein coupled receptor.AR- adrenergic receptor; ROS-reactive oxygen species.

GPCR	Gα Coupling	Mechanism of Protection in DOXTOX Models	Reference
β1-Adrenergic Receptor	Gαs	In β1-AR null primary cardiac myocytes and fibroblasts, there was decreased doxorubicin toxicity, therefore implying the cardiotoxic effects of β1-AR. β1-AR density is increased in the myocardium of DOXTOX patients. Further, β-AR activation inhibits mitochondrial complex-I that promotes cardiotoxicity.	[104,112,113,114]
Angiotensin II Receptor (ATII R1)	Gαq	α1-AR/AKAP-Lbc signaling protects cardiomyocytes against DOX-induced cardiotoxicity, both in vitro and in vivo, through the enhanced expression of the anti-apoptotic protein Bcl2, which is downregulated by DOX.	[105,106]
Substance P(NK-1R)	Gαq, Gα12, Gα13	DOX induces expression of substance P in H9c2 cardiac myocytes and mouse myocardium in vivo. Substance P antagonism reduces apoptosis and attenuates ROS production and hypertrophy in DOX-treated myocytes in vitro and in vivo.	[107,108]
Opioid Receptors (μOR-Morphine)	Gαi	In male Wistar rats, direct stimulation of myocardial δ1-opioid receptor increases intracellular ROS via activation of mitochondrial K channels, reduces cardiac function, and increases intracellular free radical signals.	[111,115]
GPR35	Gαi/o, Gα13	GPR35 mutations are associated with enhanced sensitivity to DOXTOX in pediatric patients. GPR35 overexpression decreases myocyte viability and promotes morphological changes in vitro.	[116,117]

## 7. Orphan GPCRs—New Frontiers in Pathogenic Mechanisms for Anthracycline Cardiotoxicity

Orphan GPCRs (receptors not yet associated with an endogenous ligand for activation) provide a means by which to expand the repertoire of drug targets for multiple diseases. Many orphan, or recently deorphanized, GPCRs display therapeutically relevant tissue distribution profiles and/or have links to disease [118,119]. GRP35 was previously implicated in hypertension, coronary artery disease, and heart failure [120,121]. A potential receptor for LPA and CxCL17, mutations in GPR35 are associated with enhanced sensitivity to DOX cardiotoxicity in a pediatric cohort with early-onset cardiomyopathy [117] (Table 3). The overexpression of GPR35 decreased cell viability and promoted morphological changes in myocytes in vitro [120,122]. Moreover, GPR35 overexpression confers drug resistance in NSCLC cells via β-arrestin/Akt signaling [123]. Thus, GRP35 antagonists may have a dual action in DOX cardiotoxicity by enhancing the anti-tumor and minimizing the cardiotoxic effects of DOX.

Similarly, our analysis of previously published data revealed a series of orphan GPCRs with significantly altered expression in DOX-treated iPSC-derived cardiac myocytes, which maintain disease-specific phenotypes in culture [124]. Both losses and gains of expression with DOX treatment were observed (Figure 3). Indeed, many of the downregulated receptors have paralogs that are conversely regulated. The most significantly downregulated gene, GPR26 (25-fold), has an upregulated paralog (GPR78) (Figure 3A). Both receptors are constitutively active in the absence of ligands and modulate cAMP generation [125], suggesting functional redundancy that may compensate for the loss of GPR26 (Figure 3B). Moreover, GPR78 was identified as a possible susceptibility gene through a pathway analysis of genetic defects that promote DOX cardiotoxicity [126]. Of all of the dysregulated orphan GPCRs in our analysis, 43% (decreased GPR17/22/26/34/68/132; increased GPR25/27/62/78/85/87/150) (Figure 3) are constitutively active in the absence of ligands [127]. Similarly, GPR153/GPR162 [128] and GPR27/GPR173/GPR85 [128] are all differentially regulated paralogs with likely no overall change in function in these pathways. To determine whether these GPCRs may have a significant bearing on heart physiology, we determined the relative expression levels and cell expression profiles in adult human myocardium (Table 4). The GPCRs with the highest cardiac expression (relative to other organs) were GPR22, GPR85, and GPR116, with GPR22 and GPR116 being significantly downregulated by DOX treatment. Conversely, many of the lowest expressed GPCRs (GPR25/27/113/150) were significantly increased in expression upon DOX treatment. In almost all cases, the GPCRs were expressed by cardiac myocytes, with only GPR150 showing significant cell-type restriction and GPR17/22/113 showing significant myocyte enrichment. Thus, the dysregulated GPCRs seem to participate in cardiac homeostasis.

GPR22 expression is lost from the myocardium upon DOX treatment (Figure 3). Similarly, the cardiac expression of GPR22 is lost in response to aortic banding, suggesting regulation in response to cardiac stress [129]. GPR22-null mice (despite normal cardiac development) show increased susceptibility to functional decompensation in response to elevated afterload [129]. These protective effects of GPR22 would be lost during anthracycline-induced cardiotoxicity. GPR68 and GPR132 are proton-sensing GPCRs [130] that use the protonation of the imidazole group of critical histidine residues to change hydrogen bonding, receptor conformation, and activation state. The loss of GPR68 and GPR132 attenuates the ability of myocytes to detect changes in the extracellular milieu, notably tissue acidosis, and mounts a protective response as per the response in myocardial infarction [130]. However, the role of GPR68/GPR132 may be more important in the inflammatory response where both receptors promote tumor-induced immune suppression [130]. Other dysregulated orphan GPCRs, such as the lysophosphatidylserine receptor GPR34 [131] (Figure 3B), are involved with immune function. GPR34 activation in macrophages, monocytes, dendritic, natural killer and B cells suppresses extravasation and the release of inflammatory chemokines and cytokines [132]. While the effects of GPR34 loss on myocytes are currently uncharacterized, the loss from resident cardiac monocytes/macrophages would promote inflammation and exacerbate DOX-induced injury.

A significant number of orphan GPCRs are increased in response to DOX in iPSC-CMs, indicating potential new targetable pathogenic mechanisms that mediate DOX-induced myocyte injury (Figure 3). Adhesion GPCRs and those with ECM-based ligands (increased GPR64/113/114/179; decreased GPR116/158) are overrepresented in the list of dysregulated genes. Adhesion GPCRs are characterized by long N-termini with multiple functional domains and seven highly dissimilar transmembrane regions when compared to other GPCRs. The functional domains in the N-termini promote interaction with other cells or extracellular matrixes and endow this class of GPCRs with integrin- or cadherin-like functions. Given the importance of mechano-transduction in the regulation of cardiac homeostasis (reviewed [133]), it is conceivable that changes in this class of GPCRs are important in the structural changes and myocyte apoptosis that accompany DOX-mediated cardiac injury. GPR68 is also suggested to be a mechanoreceptor [130], the activation of which may be changed by the altered expression of the adhesion GPCRs.

Increased levels of GPR50, GPR62, and GPR88 are likely cardiotoxic as they directly impact the function of one of the cardioprotective GPCRs, the melatonin receptor (MTNR1A/B). GPR88, a paralog of MTNR1B [128], competes for melatonin binding, attenuating MTNR1B activation. Through heterodimerization, GPR50 inhibits MTNR1A and GPR62 shows reciprocal modulation of MTNR1B [134]. Given the propensity of GPR62 for spontaneous β-arrestin recruitment [134], the dimers might be internalized in the absence of ligands, resulting in MTNR1A/B destruction or desensitization, an observed feature of the DOX-treated myocardium. The converse is true for apelin signaling. Both apelin and its cognate receptor are lost from the myocardium upon DOX treatment; however, GPR25 was recently identified as an apelin receptor. The upregulation and activation of GPR25 may explain why apelin infusion was shown to be protective in models of DOX-induced cardiotoxicity where APJ expression was lost [79,80].

The Drosophila ortholog of GPR155 (Anchor) suppresses BMP signaling with effects on wing size and vein patterning/size [166]. BMP2 signaling is protective in DOX cardiotoxicity by preventing myocyte death (reducing circulating Troponin T) through the Smad1 induction of Bcl-XL and preventing inflammasome formation [166,167]. Thus, the enhanced expression of GPR155 may promote myocyte apoptosis and cardiac inflammation in response to DOX through the suppression of BMP signaling.

In terms of our original premise that new medications should augment tumor killing whilst sparing the heart from cardiotoxicity, we find that antagonists of GRP85 and GRP87 potentiate the tumor-killing activity of DOX [168]. Given the increased expression in cardiac myocytes after DOX treatment (Figure 3), antagonists of GRP85 and GRP87 may represent dual-acting compounds. Moreover, yeast two-hybrid screening identified ATRX, TOP2B, and BAZ2B as direct binding partners of GPR88 [169]. The known involvement of these GPR88 interactors in chromatin remodeling suggests a direct role for this orphan receptor in DOX cardiotoxicity and a viable target for dual inhibition, although its role in cancers where DOX is used as therapy is largely undocumented. However, GPR88 may play another, more important role in DOX cardiotoxicity. GPR88 inhibits both G protein- and β-arrestin-dependent µOR signaling (this is good, as opioid receptor activation is cardiotoxic), but can also impede β-arrestin recruitment by all GPCRs in close proximity [170]. This unsuspected buffering role of GPR88 can prolong GPCR signaling and change the potency of all GPCRs in the DOX-treated myocardium where the termination of signaling is dependent upon receptor internalization, degradation, and/or recycling.

## 8. Cardiac Gαβγ Subunit Expression Is Altered by Anthracyclines

Several studies have linked changes in G protein subunit expression to a number of illnesses, including cardiovascular diseases [171,172,173], arrhythmia [174], hypertrophic cardiomyopathy [172], and heart failure [175]. Moreover, small-molecule inhibitors of βγ-signaling promote recovery and functional improvement in preclinical models of heart failure [175]. Our analysis of differentially expressed genes indicated that DOX treatment enhanced the expression of Gα14 (1.8- ± 0.44-fold) and GαL (1.73- ± 0.29-fold) in iPSC-CMs derived from patients with DOX cardiotoxicity, whilst Gα15 expression was abolished (0.0037- ± 0.017-fold) and Gαt2 was attenuated (0.5645- ± 0.1308-fold) (Figure 4A).

The impacts of GαL and Gαt2 are largely undocumented in cardiac biology; however, GαL is expressed highly in the vascular tree and Gαt2 in atrial and ventricular myocytes (*Human Heart Cell Atlas*; [176]). Conversely, the roles of Gα14 and Gα15 are better described. Gα14, along with Gα11, is important in the regulation of endothelial function and vascular development [177], with dominant expression in coronary endothelial cells in the normal heart (*Human Heart Cell Atlas*; [176]). Indeed, elevated Gα14 expression in the pulmonary vasculature is associated with pulmonary arterial hypertension [177], suggesting a role for GNA14 in vasoconstriction. The homeostatic function of Gα15 in cardiac tissue was established when SNPs in GNA15 were correlated with an average 1.54-fold increased risk of heart failure [178], likely due to its activation by cardiotoxic GPCRs such as AR, ET1, and ATIIR (see above). The mechanism may be through the regulation of inflammation as patients with GNA15 mutations experience a 6.4-fold increased risk of myocarditis when challenged with clozapine [179]. Whilst not differentially expressed in our analysis, other α-subunits (such as Gαq, Gαh) spontaneously produce heart failure in transgenic mice [180,181], suggesting that activation, not differential expression, might more accurately reflect their role in anthracycline-induced cardiomyopathy.

The regulation of cardiac pathophysiology is not solely confined to heterotrimeric α-subunits, with the contributions of β- and γ-subunits being equally important. Our analysis suggests that GNB3 (0.36 ± 0.12), GNG2 (0.36 ± 0.18), GNG7 (0.305 ± 0.17), and GNG13 (0.28 ± 0.39) are lost from myocytes and GNG4 (3.6 ± 0.99) are upregulated in response to doxorubicin (Figure 4A). Polymorphisms in GNB3, a myocyte-enriched β-subunit (*Human Heart Cell Atlas*; [176,182]), are a risk factor for postural tachycardia syndrome [183] and GNB3 is uniquely dysregulated in peripartum cardiomyopathy compared to other dilated myopathies [182], suggesting that the loss of GNB3 in DOX-induced cardiotoxicity is not a common feature of heart failure. Further, the G protein γ-subunits, GNG2 and GNG13, are amongst the hub genes upregulated in hypertrophic cardiomyopathy [184]. The enlargement of myocytes characterizes hypertrophic cardiomyopathy, while DOX cardiotoxicity is associated with myocyte apoptosis [185,186,187] and atrophy [188]. Given the opposing consequences of these heart conditions, it is logical to anticipate the downregulation of GNG2 and GNG13 in DOX cardiotoxicity. Indeed, we identified this negative correlation in the transcriptome of iPSC-cardiac myocytes from patients diagnosed with DOX cardiotoxicity (Figure 4A), suggesting loss of γ-subunits may be key to cardiac remodeling and a high value target to preserve cardiac mass during anthracycline therapy.

In terms of the importance of both βγ- and α-subunits, a series of findings over the last 20 years show that G protein activation is not just reliant on the classical GDP/GTP exchange that comes from GPCRs and GAP proteins [189]. The receptor-independent activation pathway primarily involves nucleoside diphosphate kinases (NDPKs), encoded by the nme/nm23 gene family (reviewed in [190]). Hexamers of NDPKs form a complex with Gβγ subunits and catalyze the transfer of a γ-phosphate group from an NTP to an NDP via the formation of a high-energy intermediate on the His^118^ residue of the NDPK [190,191]). The phosphate is likely transferred to the GDP dissociating from the Gα subunit, reforming GTP and triggering G protein activation in the absence of a GPCR agonist [191]. As such, NDPKs regulate the amount of GTP available for G protein activation. Further, NDPK2/3 are key regulators of G protein membrane localization in neonatal rat cardiac myocytes [192,193] and the loss of NDPK2 in zebrafish promotes the instability of G protein subunits [194], indicating that the effects of NDPKs are more complex than simple “on/off switches”. Single-nucleus transcriptional profiling identified the ubiquitous expression of NME3-7 and NME9 in the adult human heart, which was greatly enriched in myocyte populations, perhaps suggesting a role in the maintenance of heterotrimeric G protein signaling under basal conditions (*Human Heart Cell Atlas*; [176]). In heart failure, the expression of NDPK3 and the membrane content of NDPK 2 and NDPK3 are increased in myocytes [195,196,197], and NDPKs undergo substrate switching, changing preferences from Gαs to Gαi proteins, which is commensurate with the loss of cAMP signaling associated with heart failure [195]. Our transcriptional analysis of iPSC-CMs from DOX cardiotoxicity and normal patients (Figure 4B) indicates a global loss of NDPK expression upon DOX exposure, making anthracycline toxicity distinct from other forms of heart failure. However, the loss of NDPK expression may contribute to the loss of other Gαβγ subunits in anthracycline cardiotoxicity and, potentially, the altered functionality of the GRK-β arrestin signaling pathway, explored in further detail in the next section.

## 9. GRK and β-Arrestin Levels Are Altered in Doxorubicin Cardiotoxicity

GRKs and arrestins regulate multiple signaling pathways in the cell, both GPCR-initiated and receptor-independent. The two mammalian non-visual arrestins (ARRB1 and ARRB2) function as essential adaptors for the desensitization, trafficking, and signaling of GPCRs. Many GPCRs internalized through the β-arrestin–clathrin-coated pit pathway are targeted for degradation. Recently, a study of patient myocardium showed the global loss of the ubiquitination of β-ARs in patients with DOX-induced heart failure compared to other forms of heart failure [114]. These data strongly suggest that GPCR internalization and proteasomal degradation is compromised in DOX cardiomyopathy.

Our analysis of DOX-regulated genes in iPSC-derived cardiac myocytes identified significantly decreased expressions of GRKs 5/6/7 and arrestins β1 and β2 (Figure 5). The attenuation of both ARRB1 and ARRB2 expression by DOX can significantly attenuate the termination of GPCR signaling in the myocardium. β-arrestin2 acts as an essential adaptor for Nedd4 and the loss of β-arrestin expression prevents Nedd4-mediated receptor ubiquitination and lysosomal degradation [198]. Collectively, the consequence of lost β-arrestin function is the dysregulation of all heterotrimeric G protein signaling in the DOX-treated heart, which is often observed [199].

A study that identified the loss of GPCR ubiquitination in biopsies of patient myocardium determined that the expression of GRK2, GRK3, and GRK6 was unchanged in non-failing hearts versus those with anthracycline cardiotoxicity [114]. Conversely, the elevated expression of GRK5 was observed. Our analysis of iPSC-CMs found no significant change in GRK2/3 expression and a small but significant change in GRK6 levels (17 ± 3.5%), which may not be reflected in the protein content (Figure 5). Moreover, there was a significant loss of GRK5 mRNA (69.7 ± 12.2%), suggesting that changes in protein and mRNA may be dysregulated in anthracycline cardiotoxicity. In other myopathies, GRK5 phosphorylates and inhibits the cardiac mineralocorticoid receptor (MR), resulting in cardioprotection [200]. Based on these data, the increased GRK5 expression in response to anthracyclines may be an attempt to restore homeostasis though the inhibition of both GPCR and MR signaling. Indeed, differing levels of GRK5-mediated MR phosphorylation may explain the inconsistent results from the use of eplerenone in the treatment of DOX cardiotoxicity [201]. Finally, the protein analysis of patient samples [114] did not include GRK7, the most highly regulated GRK in our analysis (decreased 80%; Figure 5). In the heart, GRK7 expression is primarily confined to the ventricular myocytes and fibroblasts [202]; however, little is known of the role GRK7 plays in the myocardium.

Collectively, the above changes in GRK and arrestin expression greatly alter the compartmentalization of GPCR signaling in the myocardium (reviewed in [203]). The compartmentalization of GPCR signaling is a relatively new concept that builds on emerging evidence that many GPCRs continue to activate G proteins from intracellular compartments after they have been internalized, not just from the plasma membrane. Moreover, G protein signaling from intracellular compartments is more sustained compared to that at the plasma membrane, and its proximity to subcellular structures (such as the nucleus) radically changes the perception of how processes such as gene transcription are controlled by GPCRs (reviewed in [203]). The loss of GPCR internalization can limit GPCR signaling to the membrane when, following the detachment of β-arrestins in early endosomes, GPCRs would normally resume signaling from intracellular locations such as the nucleus (α1A-AR, α1B-AR), Golgi apparatus (β1AR), and mitochondria (CB1R, MT1R). This affects many GPCRs already implicated in modulating DOX cardiotoxicity, with many more yet to be identified.

Of the arrestin-like proteins assessed, only ARRestin Domain Containing (ARRD)C2 was upregulated by DOX treatment (twofold; Figure 5). Whilst dispensable for the ligand-induced internalization of GPCRs, ARRDC2 is required for sorting internalized receptors into endosomes and interacts with the WW-domain Nedd4 through the two PPXY motifs [198]. With the decline of β-arrestin levels (Figure 5), it seems that this function of ARRDC2 is unlikely to be important in the DOX-treated myocardium. Conversely, ARRDC2 is upregulated by stimuli that suppress growth in skeletal muscle, suggesting that this response might be part of the atrophic signaling pathway characteristic of DOX cardiotoxicity [204]. Further, ARRDC2 interacts with TopIIIb [205]. The outcome of this interaction is currently uncharacterized in the heart; however, TopIIIb is a susceptibility gene for DOX-mediated damage in genome-wide screens in yeast [206,207,208]. TopIIIb does not change the stability or translation of most target mRNAs, but stabilizes and controls the topology and translation of a subset of mRNAs [209]. These activities are both dependent and independent of topoisomerase activity, suggesting that protein–protein interactions may play a role in dictating translational outcomes [209]. As such, ARRDC2–TopIIIb complexes might modify the transcriptional landscape of the myocardium to promote cardiac myocyte atrophy, a cornerstone of the cardiotoxic phenotype of anthracyclines.

## 10. The Multifaceted Role of RGS Proteins in Doxorubicin Cardiotoxicity

Regulator of G protein-signaling (RGS) proteins play a key role in the regulation of GPCR signaling through the conserved ≈120-aa RGS region that acts as a GTPase-activating protein (GAP) for G_α_ subunits [210]. Emerging evidence demonstrates that RGS proteins are needed for normal cardiovascular function; cardiovascular abnormalities are observed in mice overexpressing RGS4 [211] or deficient for RGS2 [212], and RGS proteins are altered in various cardiovascular disease states [210,213]. Although variable results have been obtained, the expression profile of RGS proteins in failing human hearts is altered. Mittmann and colleagues [214] observed increased RGS4 mRNA but no change in RGS2 or RGS3 in such hearts, whereas Owen et al. [215] found the upregulation of RGS3 and RGS4 protein and mRNA in human heart failure, and Takeishi and colleagues [216] identified an apparent decrease in RGS2 protein.

Of the 20 RGS proteins, 6 were downregulated in iPSC-CMs by DOX treatment, whilst 3 were upregulated (Figure 6A). The R4 subfamily, notably RGS1 and RGS4, were amongst the most downregulated RGS transcripts counterbalanced by the upregulation of RGS16. RGS1 is most known for promoting inflammation in immune cells [217], so the loss of RGS1 may reduce cardiac inflammation by DOX; however, RGS1 is a key regulator of ANGII-mediated ERK activation and its loss can promote the deleterious effects of ANGII in myocytes [218]. Conversely, RGS16 is involved in the inhibition of Gα13 signaling (via a mechanism independent of GAP activity) [219]. Given the established role of Gα13-mediated signaling in exacerbating cardiac remodeling and heart failure in response to pressure overload [220], it is foreseeable that the upregulation of RGS16 is a compensatory, although futile, cardioprotective mechanism to limit cardiac damage by DOX. The R7 subfamily of RGS proteins (RGS6, RGS7, RGS9, RGS11) exist as heterodimers with the G protein β-subunit Gβ5 [221], and RGS6 and 9 are essential for the control of heart rate (and inwardly rectifying K^+^ currents) in response to the parasympathetic control of the heart [222]. The attenuation of RGS6/9 expression may explain, in part, the arrhythmogenic properties of DOX [223]. Moreover, RGS6-null mice maintain left ventricular function, heart mass, and survival 5 days after acute DOX administration (20 mg/kg), suggesting that the loss of RGS6 is cardioprotective [224]. The deletion of RGS6 reduced Ataxia Telangiectasia Mutated (ATM)/p53-induced apoptosis (decreased Bax), preserved chromosomal tail length, and dramatically reduced ROS generation [225,226,227] (Figure 7). The effects of RGS6 are independent of G protein coupling [228], reminding us that the actions of RGS proteins go far beyond the regulation of GPCRs.

The R7 subfamily of RGS proteins (both RGS6 and RGS9) have protein interaction domains (G Protein γ Subunit–Like (GGL) and Disheveled EGL-10 Pleckstrin (DEP) domains) that are absent from the remaining members. The GGL domain, a 64-residue region with a high level of similarity to the Gγ-subunit, can form dimers with particular G protein subunits (i.e., Gβ5) but not others (e.g., Gβ1 to Gβ3) [229]. This RGS/Gβ5 interaction appears to influence the protein stability [230,231], GAP activity [229,232], and subcellular localization of RGS and Gβ5 protein [233,234]. The enhancement of DOX-induced cardiac damage by RGS6 is dependent upon the 70–80-amino-acid DEP domain through poorly defined mechanisms [226].

Mice expressing the RGS9(ΔDEP) transgene fail to accumulate RGS9 in the outer segments of retinal rod cells. These data suggest that the interaction of the RGS9 DEP domain is critical for the interaction with R7/R9 anchoring proteins that alter the subcellular targeting and catalytic activity of RGS proteins [235]. As such, the loss of the RGS7 binding protein (RGS7BP; Figure 6), along with the loss of GPR158 (Fig.3), would dysregulate the locale of the remaining RGS7 subfamily members, which would have wide implications for GPCR signaling in the DOX-treated myocardium [161].

The other R7 subfamily members, RGS7 and RGS11, are significantly involved in DOX cardiomyopathy [236]. RGS11 expression is attenuated, and RGS7 expression enhanced, in the human and murine heart following chemotherapy exposure and both possess potent anti-apoptotic actions [236,237]. The action of both RGS7 and RGS11 focuses on modulating the activity of the -CaMKII-ATF3 complex. The activity of the complex is enhanced by RGS7 and attenuated by the formation of a tripartite complex between RGS11-CaMKII-ATF3 [236,237]. Not surprisingly, RGS11 overexpression and RGS7 deletion decrease DOX-induced fibrosis, myocyte apoptosis, oxidative stress, and left-ventricular dysfunction in mice [236,237]. Conversely, RGS7 overexpression and RGS11 deletion bolster the cardiotoxicity of DOX. Thus, the modulation of pro-apoptotic CaMKII signaling would seem to be a key non-G protein-related activity of RGS proteins. Further, RGS7 modulates cardiac inflammation by modulating NFκB activation and the formation of RGS7–Tip60 (acetylase) or RGS7–sirtuin 1 (deacetylase) complexes, either activating or inhibiting NFκB p65 signaling and nuclear translocation, respectively [238]. These same pathways alter cardiac fibrosis through heterotypic cell signaling between cardiac myocytes and fibroblasts [238], suggesting that the strong links between cardiac fibrosis and inflammation could be coordinated through RGS7 signaling.

Unlike the R1 and R7 subfamilies, the RZ subfamily (RGS 17, 19, 20) is mostly upregulated in response to DOX in iPSC-CMs (Figure 6). Amongst the distinguishing features of this subfamily is the N-terminal cysteine string (Figure 6), capable of post-translational modification. The palmitoylation of RGS19 promotes the association with plasma membranes [239], whilst the palmitoylation of RGS20 increases the association with, and inhibition of, Gαo [240] through both GAP activity and non-GAP mechanisms. The cysteine string region of RGS17 and RGS20 mediates an interaction with Protein kinase C interacting protein (PKCI-1) [241]. Gαz phosphorylation by PKC reduces the GAP activity of RGS proteins and prevents re-association with the βγ complex to terminate signal transduction [241]. Moreover, RGS19 acts as a scaffold that binds the E3 ubiquitin ligase GIPN through its cysteine string motif and Gα subunits through its RGS domain to promote Gα subunit degradation via the proteasome [242]. The deletion of the cysteine-rich string changes the localization from the Golgi to the nucleus in COS7 cells [243]. However, differential localization does not appear to be an important determinant for regulating G proteins. Overall, the upregulation of this class of RGS proteins is likely to result in the tighter control of Gα signaling, both through decreased GTP loading and shorter half-life, to combat the loss of class R1 and R7 family members.

## 11. Natural Compounds and the Treatment of Anthracycline Cardiotoxicity—Proof That Targeting GPCR Activity Is Effective

As previously discussed, the adverse side effects of anthracyclines often overshadow their therapeutic benefits for patients. Recently, there has been growing interest in identifying natural compounds to increase the quality of life of the patient. The combination of natural bioactive compounds and DOX has also been shown to decrease the associated side effects alongside increasing the anticancer efficacy of the drug [244]. For instance, glycyrrhetinic acid significantly protects against DOX-induced cardiotoxicity in murine models [245], possibly through blocking the activation of the GPCRs β2-AR and cysteinyl-leukotriene receptor (CysLT) [246]. Antagonists of β2-AR have already been shown to be beneficial in DOX cardiotoxicity; however, the additional benefits from the antagonism of CysLT have not yet been explored in the context of the iatrogenic effects of anthracyclines.

Both preclinical and in vitro studies have shown that DOX-induced cardiac inflammation, mediated through inflammasome pathways, can be alleviated through the use of natural compounds [247] such as curcumin (agonist of GPR97 [ubiquitous expression in heart]) [248], allicin (possible inhibitor of GRK2 [ubiquitous expression in heart]) [249], (epi)catechin (agonist for TAS2R39 [no cardiac expression] and GPER1 [ubiquitous expression in heart]) [250,251], and chrysin (GPER [ubiquitous expression in heart]) [252]. For (epi)catechin and chrysin, these results are not surprising as other pharmaceutical agonists of GPER (such as G1) are also protective in models of DOX cardiotoxicity [253]. Moreover, the established association between GRK activation and hypertrophy in DOX cardiotoxicity [254] suggests that GRK2 antagonists (like allicin) should protect the heart from anthracyclines.

Nutritional intervention with phytochemicals, as well as other natural ingredients rich in antioxidants, have shown potential in mitigating the cardiotoxic effects of anthracyclines. The polyphenolic antioxidant compound resveratrol (an agonist of adenosine receptor ADORA2A [ubiquitous expression in heart]), a major component in grapes and peanuts, is one such example [255]. Many edible plants such as mustard seeds, walnuts, and asparagus are a rich source of melatonin [256]. Whether melatonin receptor signaling acts as a possible mechanism of cardioprotective action still needs to be explored. Metformin, derived from French lilac, has been shown to inhibit the crosstalk between neurotensin receptors (NTSR1 and NTSR2 [moderate expression in heart]) and other receptors like insulin [257]. Interestingly, as listed in Table 2, targeting most of these GPCRs has already been shown to be cardioprotective in DOXTOX models using refined pharmaceutical agents or null mouse models. However, these studies provide evidence that targeting GPCR activation is a viable therapeutic approach in DOX cardiotoxicity with low to minimal side effects, as these nutritional agents are well tolerated.

However, these results from phytochemicals are yet to be proven in clinical studies. Furthermore, the lack of standardization in plant extracts remains a significant challenge in the field [256]. Hence, instead of focusing exclusively on the natural compounds themselves, leveraging well-established GPCR agonists or antagonists could provide better mechanistic clarity as well as reproducibility, helping to streamline translational applications.

## 12. Conclusions

The survival rate of cancer patients exposed to highly cardiotoxic agents has improved cancer survival but produced a “cardio-oncology epidemic” due to the iatrogenic effects of these approaches. Current FDA-approved drugs (such as dexrazoxane) produce modest clinical benefits; however, it remains uncertain which patient-specific risk factors and circumstances necessitate these cardioprotective effects. As such, better therapies and diagnostics to identify patients at risk of developing anthracycline cardiotoxicity are urgently required. The investigation of GPCR signaling in anthracycline cardiotoxicity is long overdue given their implications in other cardiac syndromes. GPCR antagonists, notably, angiotensin receptor blockers (ARBs), have shown considerable promise in addressing heart failure, proving both effective and safe in preclinical and clinical trials [258]. This review emphasizes the importance of GPCRs and their downstream regulatory mechanisms in the pathogenesis of DOX cardiotoxicity through the analysis of gene dysregulation and a comprehensive discussion of the dysregulated cardioprotective and cardiotoxic pathways. We aim to expand the list of potential targets within this established and promising therapeutic framework. The literature has highlighted how the activation of various GPCRs (such as APJ, CB1, and Wnt signaling), Gαβγ subunits, GRKs, arrestins, and RGS proteins impact the development of anthracycline cardiotoxicity. Antagonists of different GPCRs (like GRP35, GRP85, and GRP87) have shown promising results not only by reducing the DOX cardiotoxicity, but also by enhancing the anti-tumor properties of anthracyclines. As evidenced by our findings, we are optimistic that the de-orphanization of GPCRs and other pathways identified herein will open new doors for the discovery of novel therapeutic targets/agents to more effectively treat this disease. In conclusion, this review offers insight into the importance of the GPCR signaling pathway, an unexplored yet promising avenue in treating anthracycline cardiotoxicity, and we encourage researchers to build upon the findings presented herein.

## Figures and Tables

**Figure 2 cells-14-00169-f002:**
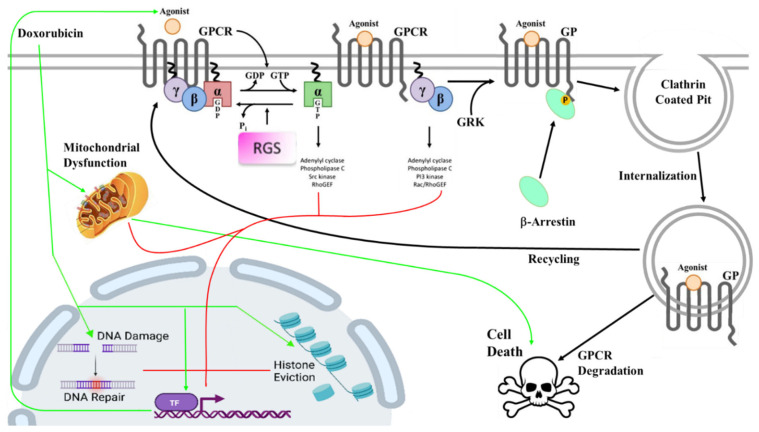
**Schematic of GPCR signaling.** Receptor activation in response to agonist promotes GTP loading of Gα subunits, signaling by Gα and Gβγ subunits, and changes in transcriptional profiles and cellular homeostasis. GPCR signaling is terminated by RGS-protein-mediated GTPase activity and by receptor phosphorylation (by GRKs), binding of β-arrestin and receptor internalization, and lysosomal targeting. The schematic also shows the targets of DOX cardiotoxicity including mitochondrial dysfunction and DNA and chromatin damage. Direct (→) and indirect (→) effects of DOX in cardiomyopathy shown.

**Figure 3 cells-14-00169-f003:**
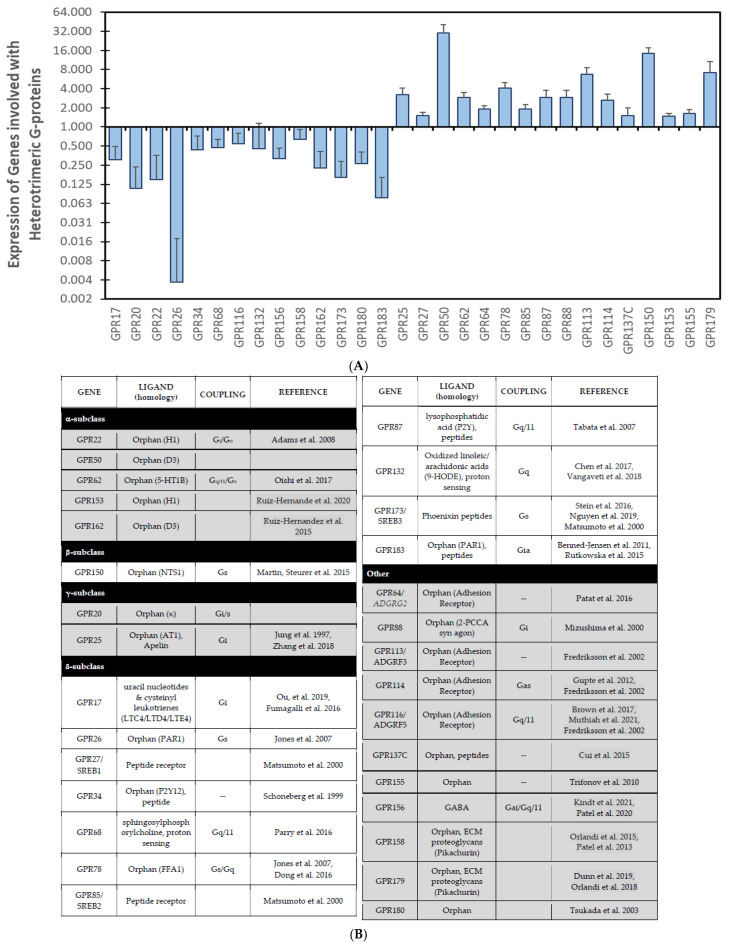
**Identification of new orphan GPCRs associated with DOX cardiotoxicity.** (**A**) iPSCs were derived from cancer patients who developed cardiotoxicity after DOX therapy or controls who did not. Transcriptional profiles of iPSC-derived cardiac myocytes (iPSC-CM) in response to DOX identified orphan GPCRs with significantly altered mRNA expression (*p* < 0.05; >1.5-fold change in expression). Data are mean ± SEM (n = 3). (**B**) Table outlining the known features of the dysregulated orphan GPCRs. GPR—G protein receptor; ADGRG—adhesion G protein-coupled receptor; ECM—extracellular matrix. References [125,126,127,128,129,130,131,132,133,134,135,136,137,138,139,140,141,142,143,144,145,146,147,148,149,150,151,152,153,154,155,156,157,158,159,160,161,162,163,164,165].

**Figure 4 cells-14-00169-f004:**
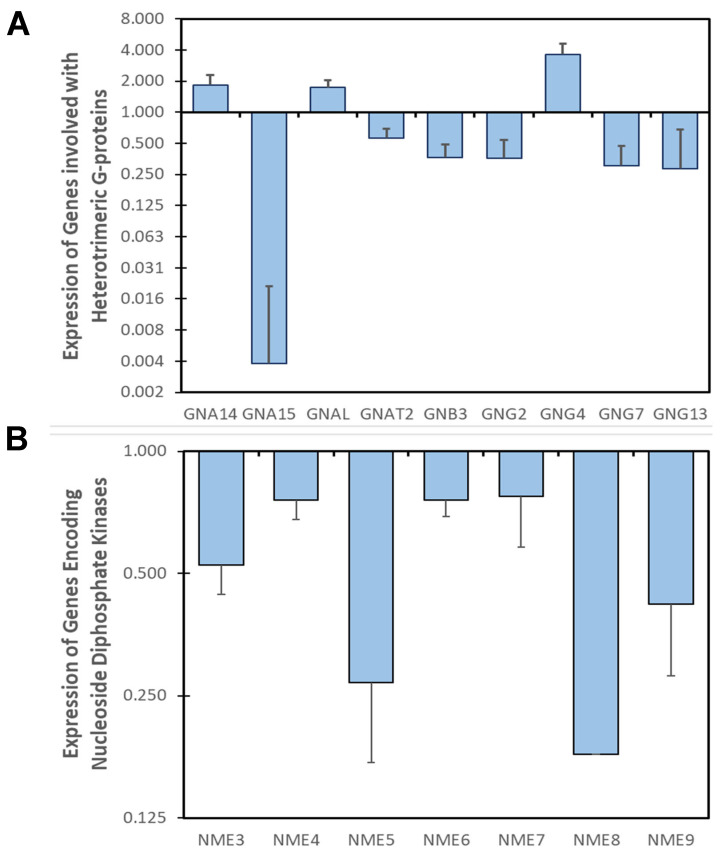
**Changes in G protein subunit and nucleoside diphosphate kinase expression in DOX cardiotoxicity.** (**A**) iPSCs were derived from cancer patients who developed cardiotoxicity after DOX therapy or controls who did not. Transcriptional profiles of iPSC-CM in response to DOX revealed a number of G protein subunits (**A**) and nucleoside diphosphate kinase enzymes (**B**) with significantly altered expression (*p* < 0.05; >1.5-fold change). Data are mean ± SEM (n = 3).

**Figure 5 cells-14-00169-f005:**
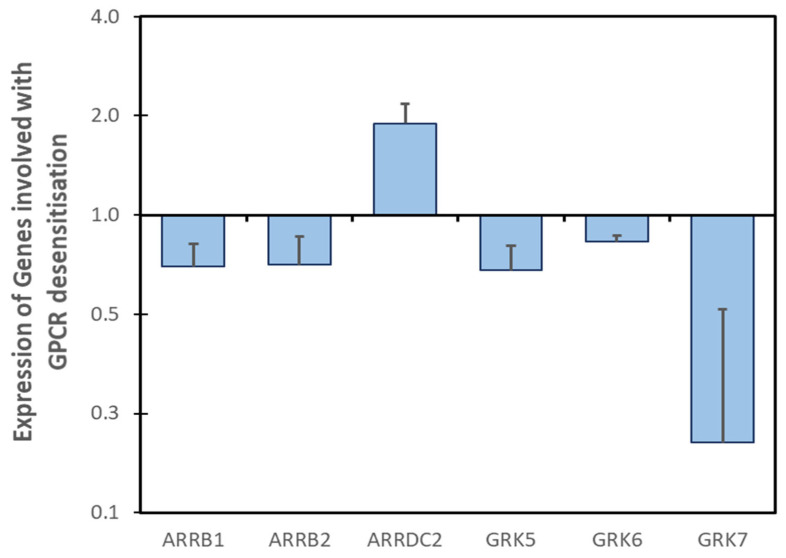
**Changes in GPCR kinase and arrestin protein expression in DOX cardiotoxicity.** Transcriptional profiles of iPSC-CM from cancer patients who developed cardiotoxicity after DOX therapy or controls who did not. DOX challenge revealed arrestin and GRK proteins with significantly altered mRNA expression (*p* < 0.05; >1.5-fold change in expression). Data are mean ± SEM (n = 3).

**Figure 6 cells-14-00169-f006:**
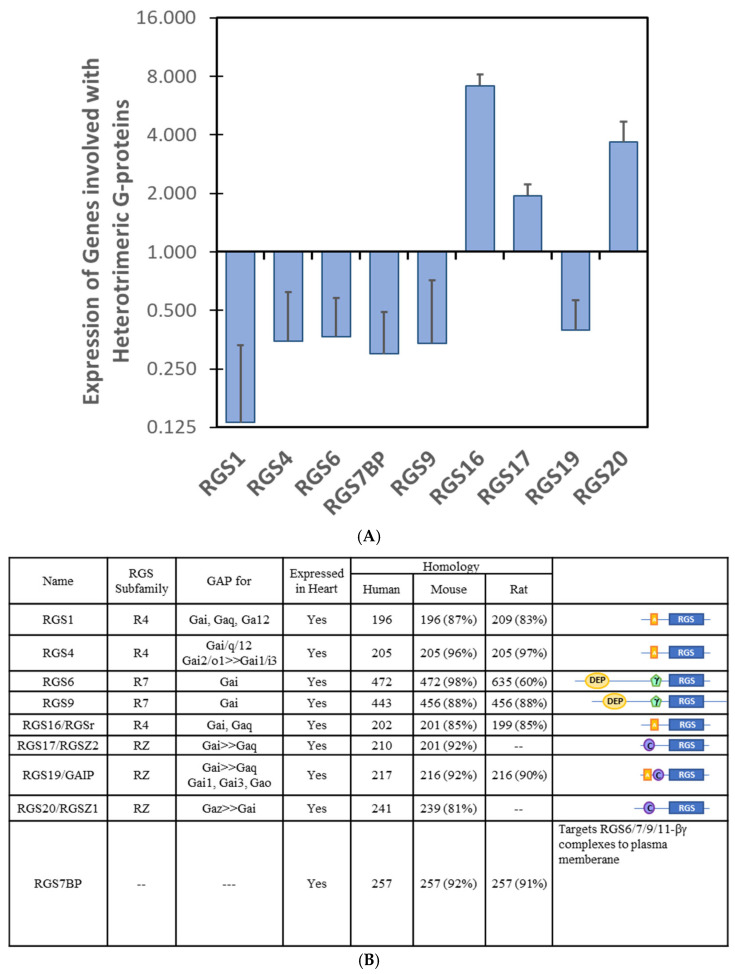
**Changes in RGS expression in DOX cardiotoxicity.** (**A**) iPSCs were derived from patients who developed cardiotoxicity after DOX therapy or controls who did not. Transcriptional profiles of iPSC-CM in response to DOX revealed a number of G protein subunits with significantly altered mRNA expression (*p* < 0.05; >1.5-fold change in expression). Data are mean ± SEM (n = 3) (**B**) Table outlining the known features of the dysregulated RGS proteins with their subfamily classification (R4, R7, RZ), G proteins targeted by their GAP activity, homology across species (percent identity compared with humans), and general schematic of composite protein domains, RGS catalytic domain; 
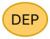
 DEP domain; 
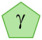
, GGL domain; 
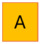
, amphipathic helix; 
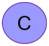
, cysteine string.

**Figure 7 cells-14-00169-f007:**
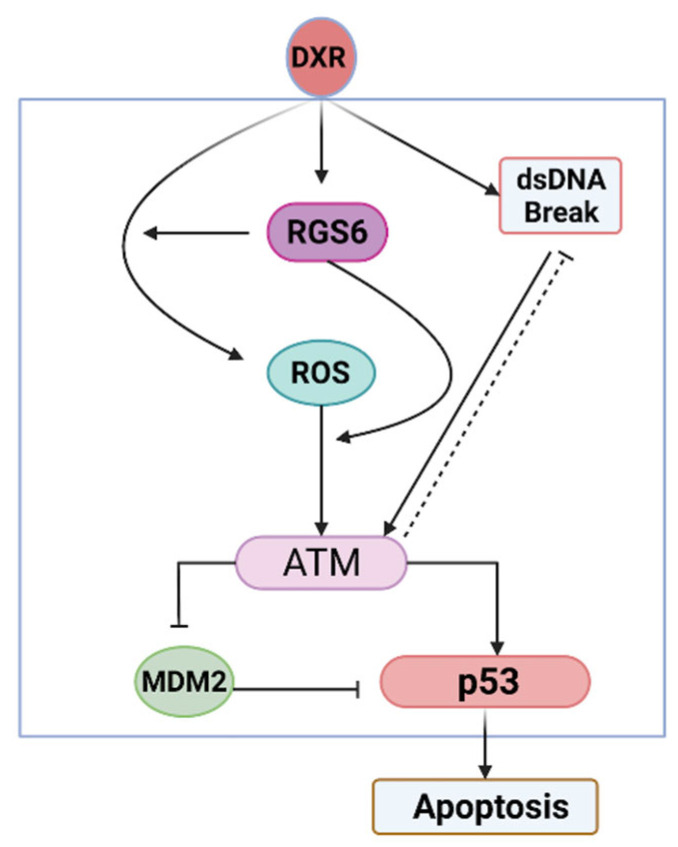
**Schematic of the roles of RGS6 in** **doxorubicin cardiotoxicity.**

**Table 4 cells-14-00169-t004:** **Expression profiling of orphan GPCRs in human myocardium.** The level of expression and cell types expressing the significantly dysregulated orphan GPCRs in our analysis were performed using the GTEx (https://gtexportal.org) and *Human Heart Cell Atlas* (https://www.heartcellatlas.org/) respectively (both accessed 18 November 2024). By comparing mRNA levels (as transcripts per million; TPM) in the adult human heart to other organs, we stratified the expression as high (■; top 30% of organs), moderate (■, 31–69%), and low (■, lowest 30%) expression. Cell type expression: A-Cm, atrial cardiac myocyte; V-Cm, ventricular cardiac myocyte; Fb, fibroblast; Endo, endothelial cell; Mural cells, vascular smooth muscle and pericytes.

GPCR	mRNA Expression (TPM) Compared to Other Organs	Prominent Cardiac Cell Type
GPR22	3.97	V-Cm
GPR85	0.67	Immune (Myeloid), A-Cm, V-Cm
GPR116/ ADGRF5	27.59	All Cells
GPR17	9.77	A-Cm, V-Cm
GPR20	0.82	Mural cell, V_Cm., Endo
GPR62	0.37	V-Cm, Endo, Mural cell
GPR64/ADGRG2	1.16	Fb, A-Cm, V-Cm, Adipocyte
GPR88	0.2	Mural cells, Immune (myeloid), Fb
GPR137C	0.82	Fb, A-Cm, V-Cm, Endo
GPR153	7.76	All Cells
GPR155	4.83	All cells
GPR162	7.51	Fb, Mural cell, Immune (Myeloid)
GPR183	9.02	V-Cm, A-Cm, Endo, Mural Cells, Immune (Myeloid, Lymphoid)
GPR25	0.005	All Cells
GPR26	0.01	Endo, Mural cells, Fb
GPR27	2.15	V-Cm, Immune (Myeloid)
GPR34	1.35	Immune (Myeloid), A-Cm, V-Cm
GPR68	1.01	Fb, V-Cm, Immune (lymphoid)
GPR113/ ADGRF3	0.06	A-Cm, V-Cm
GPR132	0.34	All Cells
GPR150	0.1	Fb
GPR156	0.12	Fb, A-Cm, V-Cm
GPR158	0.03	Fb, A-Cm, V-Cm
GPR173	3.49	Fb, A-Cm, V-Cm
GPR180	1.73	All Cells

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
