# Peer review of "Role of GPCR Signaling in Anthracycline-Induced Cardiotoxicity"

_cells, 2025, doi:10.3390/cells14030169_

Round 1
Reviewer 1 Report
Comments and Suggestions for Authors
The review by Bishwal et al., titled “Role of GPCR Signaling in Anthracycline-Induced Cardiotoxicity,” examines the involvement of G-protein coupled receptors (GPCRs) in doxorubicin-induced cardiotoxicity. Although the cardiotoxic effects of chemotherapeutic drugs on the heart and other organ systems are known, this review reframes the pathophysiology through the lens of GPCR signaling. This novel perspective is timely, as receptor-mediated pathways often present treatable targets that can be modified by existing or new drugs.
Minor Comments
- Clarification of Focus (Lines 113–115): The authors may wish to revise the explanation of the review’s focus. Although the stated emphasis is on anthracycline cardiotoxicity, the text predominantly discusses a single member of this drug class, namely doxorubicin.
- Figure 1 Legend: The figure legend is confusing, and clarification is needed.
- Terminology: "Dox" and "anthracycline" seem to be used interchangeably. Are all the data points shown in Figure 1 based on doxorubicin alone, or do they represent a combination of anthracyclines?
- Content Description: Panel B describes heart failure, Panel C discusses breast cancer, and Panel D displays odds ratios for lymphoma, rather than incidences of cardiac events. The term “cardiac event” lacks definition here. Specifically, is heart failure the only cardiac event, or are there others? Additionally, it is unclear whether doxorubicin is administered directly to heart failure patients, or if these patients develop heart failure secondary to doxorubicin treatment for other conditions.
- Orphan GPCR Section: This section provides valuable insights into GPCR roles in DoxTox models, with Figure 3 effectively summarizing the expression of GPCR transcripts.
- Protein-Level Expression: Is there evidence that the altered transcript expression correlates with changes at the protein level? Presumably, some of the animal model data in the text examine both gene and protein expression levels of GPRs to determine functional effects.
- Baseline Expression in the Heart: What is the baseline expression level of these orphan GPCRs in the heart? Are they typically lowly expressed?
- Off-Target Effects: Given the broad expression of GPCRs across various systems, is there literature addressing potential off-target effects of modulating these receptors? While targeting GPCRs could reduce cardiotoxicity, might this intervention lead to complications in other organs or even the broader cardiopulmonary system?
Author Response
Review 1
- Clarification of Focus (Lines 113–115):The authors may wish to revise the review’s focus. Although the stated emphasis is anthracycline cardiotoxicity, the primary drug discussed is doxorubicin.
We accept the reviewer’s comment but, on this occasion, disagree. The cardiotoxicity of the anthracyclines as a drug class are through the same mechanisms (both primary and secondary as outlined in lines 135-197). As such, although doxorubicin might be the “post child” for cardiotoxicity the mechanisms highlighted here are equally applicable to the other members of the family. We have amended the description of anthracycline cardiotxicity to include the relative cardiotoxicity of Epirubicin and Idarubicin (lines 121-122). As such we have left the title and focus of the review unchanged.
- Figure 1 Legend:The figure legend is confusing, and clarification is needed.
- "Dox" and "anthracycline" seem to be used interchangeably. Are all the data points shown in Figure 1 based on doxorubicin alone, or do they represent a combination of anthracyclines?
Figure 1 is primarily data from Doxorubicin. We have amended the title of the figure to reflect this.
- Panel B describes heart failure, Panel C discusses breast cancer, and Panel D displays odds ratios for lymphoma, rather than incidences of cardiac events. The term “cardiac event” lacks definition here. Specifically, is heart failure the only cardiac event, or are there others? Additionally, it is unclear whether doxorubicin is administered directly to heart failure patients, or if these patients develop heart failure secondary to doxorubicin treatment for other conditions.
We have altered the legend of figure 1. Panels B and C refer to the incidence of cardiac events (B) and heart failure (C) in a population of breast cancer patients treated with DOX. Cardiac events is now defined in the legend as “loss of EF to ≥20% or reduction in EF by 10% from baseline”. Panel D represents the odds ratio for DOX cardiotoxicity in Non-Hodgkins Lymphoma. At all times the data in figure 1 describes the DOX-induced cardiotoxicity for the specific cancer. All patients in Figure 1 were cancer patients receiving DOX for their tumor management when they developed cardiotoxicity. The legend has been altered to show this. We hope this is now clear to the reviewer.
- Orphan GPCR Section:Is there evidence that the altered transcript expression correlates with changes at the protein level? Presumably, some of the animal model data examined both gene and protein expression. What is the baseline expression level of these orphan GPCRs in the heart? Are they typically lowly expressed?
We appreciate the comments from the reviewer. In the text we already cite instances where changes in protein expression have been documented (such as for AJP expression [lines 263-264] and RGS proteins [lines634-635]). Many of the proteins have not even been explored in DOX cardiotoxicity and some don’t even have antibodies available. To answer this, and the related questions of expression level and in which cell type, we added Table 4 (page 13) that details expression levels and patterns of Orphan GPCRs in human myocardium. We hope this is sufficient to satisfy the reviewer’s curiosity.
- Off-Target Effects:Is there literature addressing potential off-target effects of modulating these GPCRs across multiple systems? We have added lines into the section on natural compounds (lines 760-762) and conclusion (lines 777-779) highlighting the safe nature of the GPCR (ant)agonists in the review. We hope this satisfies the reviewer’s comment.
Reviewer 2 Report
Comments and Suggestions for Authors
1. Regarding the mechanisms of cardiotoxicity induced by Dox, authors did not discuss about other well known contributors such as nitrosative stress, mitochondrial dysfunction and ER stress. Also, it is important to mention briefly the inflammatory response induced by dox and its cardiotoxicity.
2. Also there are some natural products that ameliorates dox-mediated cardiotoxicity. It is better to list out some of them and their protective action is mediated through GPCR signaling.
Author Response
Reviewer 2
- Regarding the mechanisms of cardiotoxicity induced by Dox, authors did not discuss about other well known contributors such as nitrosative stress, mitochondrial dysfunction and ER stress. Also, it is important to mention briefly the inflammatory response induced by dox and its cardiotoxicity.
On pages 5-6 (lines 170-197) we explore the secondary mechanisms that promote DOX cardiotoxicity including mitochondrial dysfunction, ER stress and inflammation. The manuscript cites many recent review articles that explore these “indirect” pathways by which DOX induces cardiotoxicity. As this is not the focus of this review, we believe no further information is required.
- Also there are some natural products that ameliorates dox-mediated cardiotoxicity. It is better to list out some of them and their protective action is mediated through GPCR signaling.
In response to this comment, we have added a section entitled “NATURAL COMPOUNDS AND THE TREATMENT OF ANTHRACYCLINE CARDIOTOXICITY – PROOF THAT TARGETING GPCR ACTIVITY IS EFFECTIVE”(lines 726-768). This section outlines the results of natural products using GPCR signaling as a molecular target on DOX cardiotoxicity. We state the compounds, targets of these compounds, whether these targets have already been validated using other approaches and obstacles to greater implementation of natural products therapeutically for anthracycline cardiotoxicity. We hope this new data satisfies the reviewer’s concerns.